# Asymmetric C–H Dehydrogenative Alkenylation via a Photo-induced Chiral α-Imino Radical Intermediate

Zongbin Jia[1], Liang Cheng[1], Long Zhang[1] & Sanzhong Luo [1] ✉

The direct alkenylation with simple alkenes stands out as the most ideal yet challenging strategy for obtaining high-valued desaturated alkanes. Here we present a direct asymmetric dehydrogenative $\alpha$-C(sp$^3$)-H alkenylation of carbonyls based on synergistic photoredox-cobalt-chiral primary amine catalysis under visible light. The ternary catalytic system enables the direct coupling of $\beta$-keto-carbonyls and alkenes through a cooperative radical addition-dehydrogenation process involving a chiral $\alpha$-imino radical and Co(II)-metalloradical intermediate. A catalytic H-transfer process involving nitrobenzene is engaged to quench in situ generated cobalt hydride species, ensuring a chemoselective alkenylation in good yields and high enantioselectivities.

The catalytic asymmetric $\alpha$-alkenylation of carbonyls is a strategic C-C bond-forming transformation that grants access to chiral $\alpha$-vinyl carbonyls as versatile frameworks and synthons[1,2]. There have been major advances along this line to enable stereoselective $\alpha$-alkenylation via enolate or enamine intermediates by means of transition metal or organic catalysis[3–13]. However, most of these processes necessitate the use of pre-functionalized vinyl precursors, which require activating functional groups such as halides[3–7], hypervalent iodinium[8,9] or borate[10–13] (Fig. 1, I). Simple alkenes represent the most ideal alkenylation reagents as additional activation groups can be entirely avoided and the resulting alkenylation becomes highly atom-economic and hence synthetically appealing, however, such a process remains largely unexplored[14,15]. In a few isolated cases, the reactions were limited to special alkenes[16]. To the best of our knowledge, there are few general methods to asymmetric alkenylation with alkenes, particularly in the construction of all-carbon quaternary center[17,18].

Based on the continuous exploration on oxidative enamine catalysis with chiral primary amine, we reported that the corresponding secondary enamine would undergo a facile loss of proton upon single-electron oxidation to $\alpha$-imino radical intermediate as a result of the enhanced N-H acidity at its radical cationic status[19–22]. The chiral $\alpha$-imino radical catalysis have been applied to decarboxylative alknylation[23] and dehydrogenative allylic alkylation reaction[24]. On these basis, we explored the potential of this catalysis in achieving direct asymmetric C–H alkenylation with non-functionalized alkenes[25,26]. In this article, we report a hydrogen-transfer strategy

through $\alpha$-imino radical for direct alkenylation with simple alkenes by the ternary photoredox-cobalt-chiral primary amine catalysis (Fig. 1, II). Detailed mechanistic studies have been conducted to unveil the intricate stereoscopic control mode and electron-proton-shuttle process that is indispensable in this transformation.

## Results

### Optimization for dehydrogenative alkenylation

We started with a model alkenylation reaction between ketoester 1a and styrene 2a. A preliminary attempt by the combination of chiral primary amine 4a (20 mol%), Co(dmgH$_2$)$_2$DMAPCl 5a (8 mol%) and [Ir(ppy)$_2$dtbbpy]PF$_6$ ([Ir], 2 mol%) gave the desired product 3a with only 20% yield, 80% ee and 10:1 E/Z (Condition A, Table 1, entry 1). Further optimization with the initial catalytic system didn't lead to much improvement on reactivity and enantioselectivity (Table 1, entries 2–4). Interestingly, a slightly improved enantioselectivity was observed with the combination of 4,4-dimethylaminopyridine (DMAP) and a difluoroborane (BF$_2$)- Co(II)-catalyst 5b (Table 1, entry 5, 20% yield and 86% ee). Further investigation showed that the introduction of 2-nitrotoluene (25 mol%) as H-acceptor and decreasing the reaction temperature to −10 °C were particularly effective to facilitate the alkenylation pathway. In this process, 2-nitrotoluene was fully reduced to its aniline derivative with 56% yield (in terms of 2-nitrotoluene, supplementary Fig. 1). It was found a larger ratio of 2a/1a led to higher yield of the product 3a, while slightly reduced results for both yield and stereoselectivity were observed when excess amount of 1a was

[1]Center of Basic Molecular Science, Department of Chemistry, Tsinghua University, Beijing 100084, China. ✉e-mail: luosz@tsinghua.edu.cn

**Fig. 1 | Strategies for asymmetric α-alkenylation of carbonyls. I** Traditional alkenylation strategy with activated precursors. **II** Synergistic photoredox-cobalt-chiral primary amine catalysis for direct alkenylation in this work.

engaged (Table 1, entries 7–8). The screening of amino-catalysts revealed that the morpholine-substituted **4a** was the optimal one, and switching to piperidine **4b** and diethylamino- **4c** led to a diminishing yield and enantioselectivity (Table 1, entries 9 and 10). The use of other nitro-arenes showed comparable results (Table 1, entries 11 and 12). Under optimized conditions **B**, the desired alkenylation product **3a** was obtained in 71% isolated yield, 93% *ee,* and 18:1 *E/Z* ratio (Table 1, entry 6, Condition B). Finally, control experiments revealed that any of the catalytic system was essential in the reaction, and no reaction were observed in their absence (Table 1, entry 13). The reaction also did not proceed in the dark without light irradiation (Table 1, entry 14), verifying its photochemical nature.

## Substrate Scope

As shown in Fig. 2, styrenes bearing various *para*-substituents on the aryl ring such as alkyl (**3c**), phenyl (**3d**), alkoxyl (**3e**), phenoxyl (**3f**) and halogen (**3h–3j**) were well tolerated in the reactions to afford the *E*-selective alkenylation products with yields ranging from 30% to 74% and high levels of enantioselectivities (Fig. 2, entries 2–11). Interestingly, the configuration of alkenes was mostly of *Z*-form when a fluoro-substituted photoredox catalyst [Ir]-dF was used, and similar results were also observed for other substrates (Fig. 2, entries 1–4). The substrate with long linear alkoxyl group also worked well, providing the corresponding product **3k** with 42% yield, 90% *ee,* and 3:1 *E/Z* ratio. Moreover, *meta*-, *ortho*- as well as multi-substituted styrenes reacted smoothly with satisfying results (Fig. 2, entries 13–22). Generally, the reaction favors electron-donating styrenes (**3e–3g, 3q, 3r,** and **3w**) and slightly decreased yields were observed for styrenes bearing electron-withdrawing group (**3i, 3j,** and **3n**).

It should be noted that *ortho*- group has little effect on reactivity as 2,4,6-trimethylstyrene gave the corresponding product **3v** with 40% yield, 86% *ee,* and 6:1 *E/Z* ratio (Fig. 2, entry 23). Thiophene-substituted alkenes worked well in the reaction with single *E*-alkene stereoisomer **3w** (Fig. 2, entry 24). Unfortunately, internal *β*-methyl styrene, cyclohexane, and terminal substrates such as allylbenzene and 1,1-diphenyl ethylene did not work under the present conditions (Fig. 2, entries 41–44).

The applicability of the *β*-ketocarbonyls was next investigated. Different esters could be incorporated to furnish the corresponding alkenylation products with good yields and enantioselectivities (Fig. 2, entries 25 and 26). Dihydrofuranone, cyclohexanone or acyclic ketoesters could also be applied, showing unfortunately low reactivity (Fig. 2, entries 27–29). The reactions worked well with *β*-ketoamides to

give the corresponding single *E*-alkenylation products **3ac–3af** and the five-membered cyclopentanones showed higher activity and stereoselectivity than their six-membered counterparts (Fig. 2, entries 30–33), likely due to the more propensity of five-membered rings to form exocyclic double bond, a preferred geometry for the key radical intermediate (Fig. 1).

The current catalytic protocol could be extended to late-stage functionalization of structurally complex substrates bearing natural products and pharmaceuticals. Firstly, celestolide **3ag** and tonalid **3ah** derived alkenes showed excellent enantioselectivities and *E/Z* ratio (Fig. 2, entries 34 and 36). Furthermore, similar results were also observed for substrates bearing pharmaceutically active ibuprofen **3aj** and camphanic acid group **3aj** (Fig. 2, entries 36 and 37). Of further significance is the observation that the protocol enables late-stage functionalization of L-tyrosine and L-phenylalanine derivatives (**3ak**, **3al**) in good yields and high levels of stereoselectivity (Fig. 2, entries 38 and 39). In addition, diacetone-fructose derived *β*-ketoester also worked smoothly to furnish the corresponding product **3am** albeit with relatively low activity (Fig. 2, entry 40). Finally, a gram-scale reaction (10 mmol) of *β*-ketoester **1a** and styrene was performed to probe the practicability, and comparable results were obtained in the presence of a reduced catalyst loading (Fig. 2, entry 2).

Interestingly, 1,1 di-substituent alkenes could be also applied to this asymmetric dehydrogenative transformation. Unexpectedly, thermodynamically-stable allyl alkylation product **3an** was obtained with satisfactory result from 1-methylene-dihydroindene **2an**, an inactive substrate by our previous strategy (Fig. 3, **3an**)[11,24]. When there is no 2-nitrotoluene, only the isomerization product **2an'** is obtained with a yield of 95%, and no desired allylic adduct was isolated. Similar allylic alkylation were also obtained with five, six, and seven-membered cyclic methylenes with liner and cyclic *β*-ketoester (Fig. 3, **3ao–3as**).

## Mechanistic studies

**Acid-base effect on the catalytic system.** In asymmetric dehydrogenative allylic alkylation reaction, we noticed that balancing basicity between tertiary amino moiety of aminocatalyst and axial ligand of cobaloxime was critical for the transformation[24]. In the current reaction system with BF$_2$-Co(II) **5b**, the catalytic amount of base additive was also found to be critical for both reactivity and stereoselectivity, and there was virtually no reactivity in its absence. A survey of different organic base revealed a clear trend between basicity (p$K_{aH}$) and catalytic performance (Fig. 4, III). DMAP with a p$K_{aH}$ = 7.9 gave the best results in terms of both yield and enantioselectivity. Strong basic amines such as DABCO (p$K_{aH}$ 9.1), guanidine (p$K_{aH}$ 13.2), DBU (p$K_{aH}$ 13.9), and DBN (p$K_{aH}$ 15.3) demonstrated poor activity but maintaining selectivity. Similar behaviors were also observed with less basic amines such as *N*-methyl imidazole (p$K_{aH}$ 6.4) and pyridine (p$K_{aH}$ 3.4)[27,28]. From Fig. 4, it is also clear that basicity seems only affect activity but not the stereoselectivity, suggesting the base-mediated proton shuttle may facilitate the conversion, but do not directly participate in the stereodetermining step. In this regard, the loading of DMAP also has a dramatic effect on the catalysis, both decreasing and increasing the loading led to a reduction of activity (Fig. 4, I). Furthermore, the basicity of tertiary amine moiety also significantly influences this reaction. Piperidine and diethylamine, with stronger basicity than morpholine, markedly diminish both reactivity and enantioselectivity, highlighting a delicate acid-base balance in the reaction (Fig. 4, **II**, and **III**).

A proton-shuttle network involving the key intermediate (e.g. **6a-I**) and external bases can be invoked to account for the observed acid-base effect (Fig. 4, **IV**). DMAP (p$K_{aH}$ = 7.9) with moderate basicity would facilitate the photo-induced electron transfer by proton abstraction to form the most active radical intermediate **6a-I** as a mono-protonated species (p$K_a$ = 9.9). A stronger base may further

**Table 1 | Optimization for asymmetric dehydrogenative alkenylation.**[a]

| Entry | Variation from condition A | Yield (%)[b] | Ee/(%)[c] | E/Z |
|---|---|---|---|---|
| 1 | none | 20 | 80 | 10:1 |
| 2 | PhNO₂ as H-acceptor | 6 | 70 | >20:1 |
| 3 | –10 °C | 10 | 74 | 8:1 |
| 4 | **5b** instead of **5a** | 8 | 43 | 4:1 |
| 5 | **5b** + DMAP | 20 | 86 | 2:1 |
| | **Variation from condition B** | | | |
| 6 | none | 73(71)[d] | 93 | 18:1 |
| 7 | **1a** : **2a** = 1 : 3 | 62 | 91 | 14:1 |
| 8 | **1a** : **2a** = 2 : 1 | 40 | 85 | 11:1 |
| 9 | **4b** instead of **4a** | 20 | 81 | 19:1 |
| 10 | **4c** instead of **4a** | 13 | 72 | >20:1 |
| 11 | PhNO₂ as H-acceptor | 72 | 91 | 13:1 |
| 12 | C₆F₅NO₂ as H-acceptor | 74 | 91 | 15:1 |
| 13 | w/o **4a** or **5a** or [Ir] | <5 | n.d. | n.d. |
| 14 | In the dark | <5 | n.d. | n.d. |

H-acceptor hydrogen acceptor, *[Ir]* Ir(ppy)₂dtbbpyPF₆, *ppy* 2-phenyl-4-pyridine, *dtbbpy* 4,4'-ditert-butyl-2,2'-bipyridine, *n.d.* not detected, *w/o* without.

[a]Reaction conditions: **1a** (0.1 mmol), **2a** (0.5 mmol), **4a** (20 mol%), **5b** (8 mol%), DMAP (8 mol%), 2-nitrotoluene (25 mol%), [Ir] (2 mol%), 0.3 mL MeCN, deaerated and irradiated 48 h by blue LED under –10 °C.

[b]The yield and *E/Z* ratio were determined by GC analysis.

[c]Enantioselectivity was determined by chiral HPLC analysis.

[d]Isolated yield.

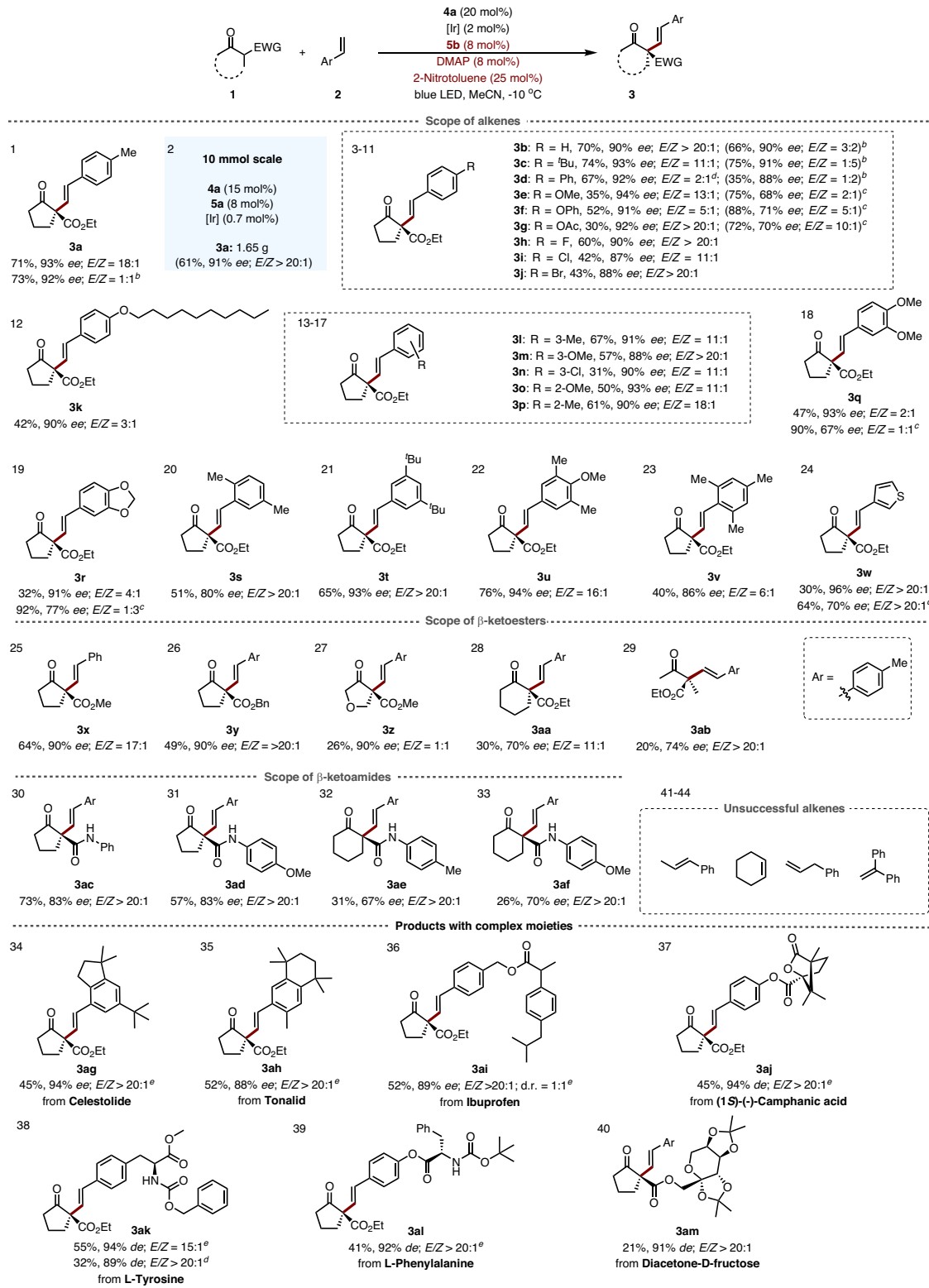

**Fig. 2 | Scope of alkenylation.** [a]Reaction conditions: **1** (0.1 mmol), **2** (0.5 mmol), **4a** (20 mol%), [Ir] (2 mol%), **5b** (8 mol%), 2-nitrotoluene (25 mol%), DMAP (8 mol%), 0.3 mL of MeCN, deaerated and irradiated for 48 h by 30 W blue LED under −10 °C. Yield with isolated product. E/Z ratio was determined by [1]H NMR analysis. ee was determined by HPLC analysis. [b][Ir]-dF instead of [Ir]. [c]Reaction under room temperature. [d]1.0 Equivalent of alkenes and 2.0 equivalent of **1a** were added. [e]2.5 Equivalent of alkenes was added.

deprotonate to form neutral radical species **6a-III**, which is less reactive in radical addition. The reactivity bias toward electron-rich styrene is in line with this scenario. In addition, the morpholine (p$K_{aH}$ = 9.2) side chain as in **4a** also favors an internal proton shift toward **6a-I**

(p$K_a$ = 9.9) over **6a-II**. On the other hand, piperidine (p$K_{aH}$ = 10.5) and diethylamine (p$K_{aH}$ = 10.9) side chains (as in **4b** and **4c**, respectively) would favor the equilibration to **6a-II**, explaining the observed poor activity of these two catalysts.

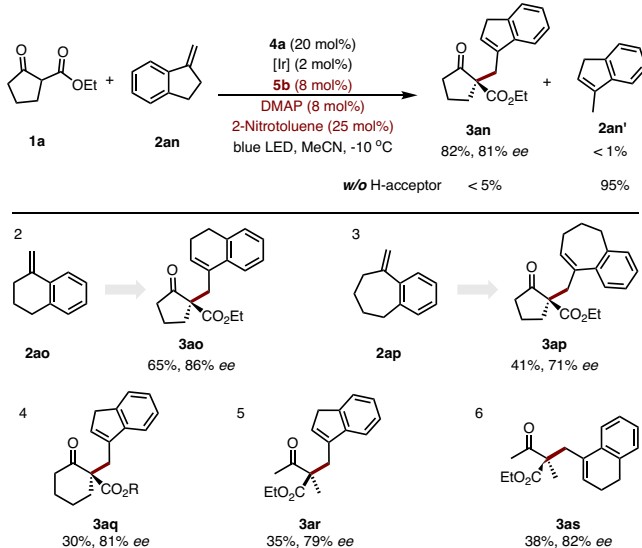

**Fig. 3 | Dehydrogenative allylic alkylation.** Reactions were performed on 0.1 mmol scale, yield with isolated product.

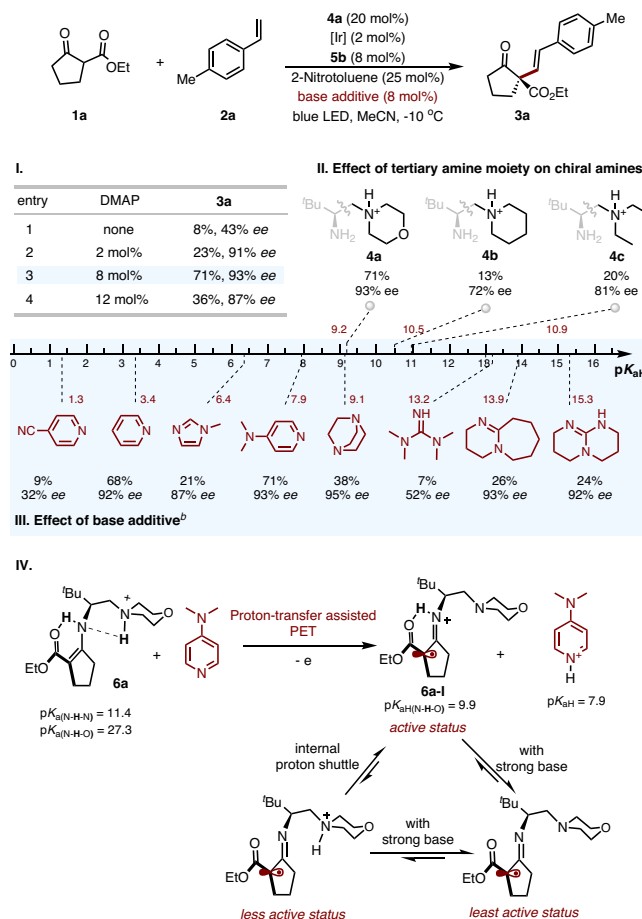

**Fig. 4 | Acid-base effect on the catalytic system. I** Effect of DMAP loading. **II** Effect of tertiary amine moiety on chiral amine. **III** Effect of base additive. **IV** Proposed proton shuttle. *a*DMAP (8 mol%) as base additive. *b*4a was engaged as catalyst, the pK_a values were recorded from the iBonD databank or determined through DFT calculation in DMSO solution.

**Fig. 5 | [Co^III]-H mediate hydrogen transfer. I** Role of hydrogen acceptor. **II** Effects of different cobalt catalysts. *a*5a as cobalt catalyst.

## Hydrogen transfer with cobalt

[Co^III]-H species are known to undergo reversible addition-elimination with alkenes[24,29]. Under hydrogen-evolving conditions in the absence of H-acceptor, the reaction afforded mainly alkene-dimerization byproduct **b1** (42% yield) (Fig. 5, I), derived from [Co^III]-H mediated radical process. The desired alkenylation adduct **3a** was isolated in a minor 20% yield and 80% *ee*. During further optimization, the use of BF_2 Co(II)-catalyst **5b** in the presence of nitrotoluene was identified to effectively suppress the hydroalkylation by-pathway, leading exclusively to the desired alkenylation reaction (Supplementary Fig. 1 vs Fig. 2). Previously, we found oxime-Co(III) catalyst **5a** was able to promote the deuteration of styrene through [Co^III]-H mediated hydroalkylation-dehydrogenation process[29]. In contrast, **5b** showed virtually no activity in the H/D exchange reaction (Fig. 5, II). On the other hand, both **5a** and **5b** showed comparable activity in the photo-reduction of 2-nitrotoluene[30–32]. These observations suggest that [Co^III]-H derived from **5b** can preferentially react with polar nitro- moiety instead of alkene, explaining the observed chemoselectivity.

## Photoredox cycle and stereocontrol model

A series of stoichiometric experiments with preformed enamine **6a** were tested under the dual photoredox and cobalt catalytic conditions. The alkenylation reaction proceeds to give the desired product **3a** with excellent enantioselectivity, verifying the enamine catalysis nature (Fig. 6, I). The addition of water to the stochiometric reaction led to a minor yet noticeable drop of the enantioselectivity. Similar detrimental polar effects were observed during the process of optimization of solvents, with protic solvents showing rather poor reactivity and enantioselectivity (Supplementary Table 8). Stepwise multivariant linear free energy correlation (LFER) analysis revealed an excellent correlation between enantioselectivity and the solvent acidity scale, Catalán's SA (Fig. 6, II)[33,34]. The observed water additive effect and solvent LFER analysis suggest a critical ionic interaction in the stereocontrolling step. Furthermore, Stern-Volmer fluorescence quenching experiments

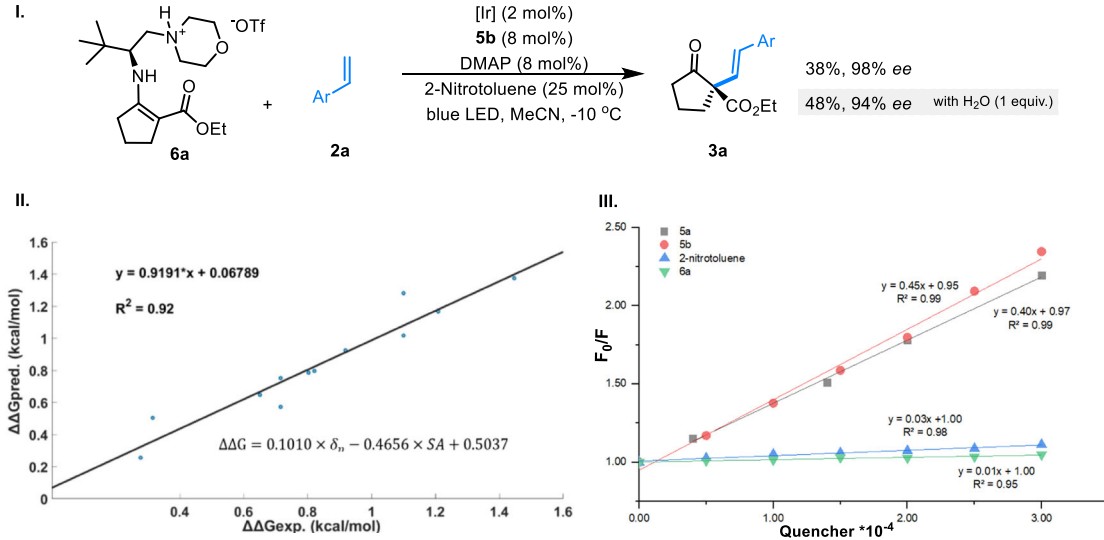

**Fig. 6 | Mechanistic investigation. I** Stoichiometric experiment with preformed enamine **6a**. **II** LFER analysis with different solvents. **III** Stern-Volmer quenching experiments with [Ir].

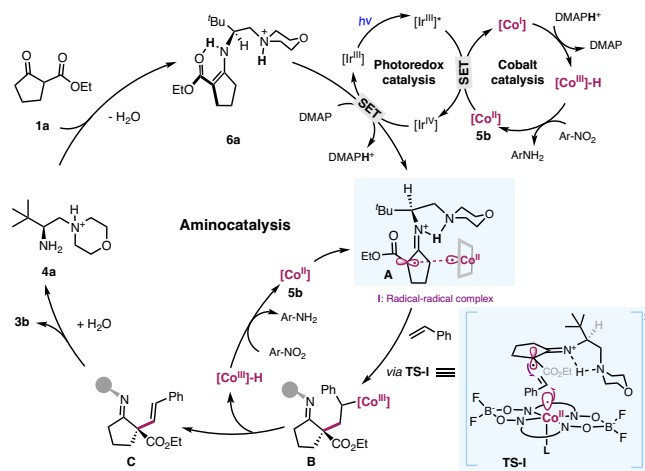

**Fig. 7 | Proposed catalytic cycle.** Synergistic photoredox-cobalt-chiral primary amine catalysis.

revealed that the excited state [Ir^III]* was only quenched effectively by cobalt **5a** or **5b**, a clear indication of a reductive quenching mechanism, supporting a SET sequence involving cobalt-iridium-enamine (Fig. 6, **III**).

**Proposed cycle.** On these bases, a catalytic cycle was proposed as shown in Fig. 7. The combination of [Ir], DMAP, and cobalt **5b** provides an efficient photoredox oxidative system, leading to the generation of α-imino radical and [CoIII]-H species through a sequence of electron and proton transfer process. A radical-radical complex **A**, an ion pair consisting of the imino radical **6a-I** and Co(II) that is sensitive to polar media, was proposed to dictate the stereoselectivity. Subsequent cooperative radical addition to alkene through transition state **TS-I** results in the formation of the critical C-C bond (Fig. 7). A following photo-mediated dehydrogenation leads to the alkenylation product and another [CoIII]-H species. Subsequent hydrolysis would regenerate aminocatalyst **4a** and complete the catalytic cycle. The reduction process between 2-nitrotoluene and the two molecules of [CoIII]-H species complete

the cobalt catalysis, and its effectiveness is critical to override the undesired hydroalkylation process.

## Discussion

We have developed an efficient catalytic system to achieve the direct oxidation of secondary enamine intermediate which can be applied to enantioselective α-C(sp3)-H functionalization of carbonyls with alkenes by combining photoredox-cobalt-chiral primary amine catalysis under visible light irradiation. This synergistic system leads to the formation of alkenylation adducts with excellent stereoselectivity through a cooperative radical addition process that involves a chiral α-imino radical and Co(II)-metalloradical. A series of mechanistic studies revealed an elaborate electron and proton transfer process that was involved. The successful development of asymmetric alkenylation is also attributed to the presence of hydrogen acceptor to quench the in-situ generated cobalt hydride, thus improving chemoselectivity. We believe the current strategy would find broad applications in elusive asymmetric radical transformations.

## Methods

### General procedure for dehydrogenative alkenylation

In an oven-dried 5 mL pyrex tube equipped with a magnetic stir bar, β-ketocarbonyls **1** (0.1 mmol), [Ir(ppy)₂dtbbpy]PF₆ (1.86 mg, 2 mol%), Co(dmgBF₂)₂ .2H₂O (3.36 mg, 8 mol%), chiral primary amine **4a** (6.72 mg, 20 mol%), DMAP (0.98 mg, 8 mol%), alkene **2** (0.5 mmol) and MeCN (0.3 mL) were added. The mixture was equipped with a rubber septum and bubbled with argon gas. The sample was then irradiated by a 30 W blue LED under −10 °C condition for 48 h. After that, the reaction mixture was directly loaded onto silica gel column and eluted with ethyl acetate/hexane to obtain the alkenylation product.

## Data availability

All data are available from the corresponding author upon request. Supplementary Information is available and includes general information, substrate and reagent synthesis, optimization details, general experimental procedures, and compound characterization, determination of the absolute configuration, mechanistic studies, HPLC, NMR spectra, and DFT calculations. Source data are present. Source data are provided with this paper.

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

## Acknowledgements

We thank the National Key R&D Program of China (2023YFA1506401, S.L. and L.Z.), the National Fundamental Resource Investigation Program of China (2018FY201200, S.L.), the Natural Science Foundation of China (22031006 and 22393891, to S.L.), and Haihe Laboratory of Sustainable Chemical Transformations for financial support (S.L. and L.Z.).

## Author contributions

S.L. conceived and directed the project.; Z.J. optimized the reaction conditions, examined the substrate scope, and studied the mechanism with the help of L.C.; L.Z. performed the computational studies. S.L. and Z.J. wrote the manuscript, with contributions from all authors.

## Competing interests

The authors declare no competing interests.
