## [Peer Review File · Nature Communications]

Asymmetric C–H Dehydrogenative Alkenylation via a Photo-induced Chiral α -Imino Radical IntermediateREVIEWER COMMENTS

Reviewer #1 (Remarks to the Author):

In this manuscript, Luo and coworkers report an unprecedented asymmetric dehydrogenative α -C(sp³)-H alkenylation by the synergistic photoredox/cobalt catalysis. A series of control experiments were performed to probe the mechanism. Overall, this is interesting work, and the reviewer believes the paper will be suitable for publication in Nature Communication once the following comments have been addressed.

1. This catalytic protocol was well tolerated with five-member ring-derived ketoesters, which showed excellent enantioselectivity. However, other ketoesters like substrate 3z, 3aa, and 3ab afforded lower enantioselectivities. Please comment.
2. For α -branched styrenes, dehydrogenative α -C(sp³)-H allylic alkylation will occur. The major difference in the mechanisms between alkenylation and allylic alkylation is that the Co(II)-mediated hydrogen atom transfer process with alkyl radical intermediate at different positions. Substrate 1-methylene-dihydroindene 2an afforded excellent regioselectivity in this catalytic protocol. What about α -methyl styrene?

Reviewer #2 (Remarks to the Author):

Complemented by enolate or enamine chemistry, the α -imino radical offers an alternative approach for the transformation of aldehydes or ketones. In 2017, the MacMillan group employed this strategy to establish direct enantioselective alkylation of aldehydes with simple alkenes through a synergistic merger of photoredox, enamine, and HAT catalysis. In 2022, the Luo group reported an asymmetric C(sp³)-H allylic alkylation of keto carbonyls via the enamine/photoredox /cobalt tri-catalytic system.

In this manuscript, Luo and colleagues present a new approach for the direct asymmetric dehydrogenative α -C(sp³)-H alkenylation of β -keto-carbonyls with simple alkenes. This transformation is achieved through synergistic photoredox-cobalt-chiral primary amine catalysis under visible light irradiation. The developed methodology provides an efficient and straightforward route to access chiral α -vinyl carbonyl compounds. Furthermore, a comprehensive set of mechanistic studies has been conducted to elucidate the acid-base effect on the catalytic system, hydrogen transfer with cobalt, photoredox cycle, and stereocontrol model. These investigations support the proposed mechanism involving photoredox-cobalt-chiral primary amine catalysis. Overall, this work represents a significant advancement in enantioselective functionalization of C(sp³)-H bonds using photoredox catalysis. Therefore, I recommend publishing this manuscript in Nature Communications after addressing the following concerns:

- 1) Cite references: i) MacMillan et al., Nat. Chem. 2017, 9 (11), 1073–1077. A latest perspective of photocatalyzed Enantioselective C(sp³)-H Functionalization: ii) Xu et al., J. Am. Chem. Soc. 2024, 146, 1209–1223.
- 2) An abbreviation “w/o” was used in table 1 and tables of SI, but the authors didn't explain its meaning in the note, hence an annotation of this abbreviation should be added in manuscript and SI.
- 3) Could you please confirm if the coupling constants for 3r-E and 3r-Z are correct?
- 4) Is there a relationship between retention time and substituents in HPLC analysis? Why do some compounds produce the peak of the racemate before, while others produce it after? For example, compounds 3s and 3t. Can you explain this?

Reviewer #3 (Remarks to the Author):

The authors have developed a new asymmetric strategy for the vinylation of 1,3-dicarbonyl compounds that merges organocatalysis, photoredox catalysis, and cobalt catalysis. This is clearly an extensive of their previous work (JACS 2022, 144, 10705) that uses a similar ternary catalytic system for the allylation of β -ketocarbonyl compounds. The major difference between this work and the previous work is the selection of alkene. The previous work used α -methylstyrenes whereas this work mostly uses styrenes that lack this methyl group, which ensures that only one elimination pathway is feasible to form the desired products. The substrate scope and limitations have been thoroughly examined and we commend the authors for demonstrating the robustness of their protocol by running the reaction on a gram scale. The mechanistic studies provide some additional insight but again there is some overlap with previous disclosures. Overall, I do not support publication in Nature Communications as this work does not significantly advance upon previous work disclosed by the same authors.

Reviewer 1:

Q1. This catalytic protocol was well tolerated with five-member ring-derivated ketoesters, which showed excellent enantioselectivity. However, other ketoesters like substrate 3z, 3aa, and 3ab afforded lower enantioselectivities. Please comment.

Reply: This is indeed an interesting phenomenon. Before that, please allow me to correct a mistake caused by our carelessness. The *ee* value of **3z-E** and **3z-Z** should be 90% (SI, page 140). As shown in Fig. 2, five-membered cyclic substrates possess higher reactivity and enantioselectivity compared with six-membered cyclic and liner substrates. We attribute this fact to differences in activity of different α -imino intermediates. Specifically, since five-membered rings are more likely to form exocyclic double bonds, for five-membered ring substrates, imino radicals are more likely to be distributed at the carbon center, while six-membered rings and linear substrates are more likely to be distributed at the N center, and thus leading to differences in reactivity (picture below, left column). Similar effect can also be observed in our previous work (JACS 2022, 144, 10705). On the other hand, for normal enamine, six-membered cyclic enamine is easily formed and possess higher activity when reaction with electrophiles (picture below, right column, refer to our previous works: Angew. Chem. Int. Ed. 2020, 59, 14347; Angew. Chem. Int. Ed. 2021, 60, 10971.).

Moreover, the steric effect on keto-ester has a noticeable impact on enantioselectivity due to cooperative radical addition mechanism. The five-membered substrates possess lower steric hindrance and conformational flexibility, making it easier to obtain high enantioselectivity.

A brief comment was added in the revised manuscript as the following: “--- likely due to the more propensity of five-membered rings to form exocyclic double bond, a preferred geometry for the key radical intermediate (Fig. 1).”

Q2. For α -branched styrenes, dehydrogenative α -C(sp³)-H allylic alkylation will occur. The major difference in the mechanisms between alkenylation and allylic alkylation is that the Co(II)-mediated

hydrogen atom transfer process with alkyl radical intermediate at different positions. Substrate 1-methylene-dihydroindene 2an afforded excellent regioselectivity in this catalytic protocol. What about α -methyl styrene.

Reply: Thanks for the suggestion. We performed this reaction with cobalt 5a as catalyst and the results showed that α -methyl styrene is well tolerated to give allylic alkylation product with high enantioselectivity. Compared with previous catalytic system, the current ternary H-transfer system enables the reaction with alkyl-substituted styrenes (Fig. 3) that are not possible in previous study (JACS 2022, 144, 10705).

Reviewer 2:

Q1. Cite references: i) MacMillan et al., *Nat. Chem.* 2017, 9 (11), 1073–1077. A lasted perspective of photocatalyzed Enantioselective C(sp³)–H Functionalization: ii) Xu et al., *J. Am. Chem. Soc.* 2024, 146, 1209–1223

Reply: Thanks for your kind suggestion. The two articles are very constructive works in this field and we are pleasure to include them as ref 25 and ref 26.

Q2. An abbreviation “w/o” was used in table 1 and tables of SI, but the authors didn’t explain its meaning in the note, hence an annotation of this abbreviation should be added in manuscript and SI.

Reply: Thanks. We added “w/o = without” to the footnote to explain it.

Q3. Could you please confirm if the coupling constants for 3r-E and 3r-Z are correct?

Reply: Many thanks for your reminder. We reversed the order of **3r-Z** and **3r-E** in the Characterization of Compounds section (SI, page 43), this has been corrected. Furthermore, for **3r-Z**, the “ δ 6.75 (d, $J = 7.8$ Hz, 3H)” have been corrected as “ δ 6.76 (s, 1H), 6.74 (s, 2H)”

Q4. Is there a relationship between retention time and substituents in HPLC analysis? Why do some compounds produce the peak of the racemate before, while others produce it after? For example, compounds 3s and 3t. Can you explain this?

Reply: Yes, it's an interesting phenomenon. We believe that the absolute configuration of all products is consistent. As we known, chiral chromatography columns separate chiral molecule through interactions with internally loaded chiral polysaccharides. Slightly change the substituent on the products can significantly affect the interaction with the chiral filler and thus change the order of the peaks, which means that the order of the peaks is not directly related to the absolute configuration. This phenomenon is quite common with chiral molecules. For two recent examples in methodology studies, Benjamin List, *Science* 382, 325-329 (2023), (6f vs 6g, SI, page 142-143); Zhiwei Zuo, *Science* 382, 458-464 (2023), (10 vs 11, SI page 84; 20 vs 21, page 87-88).

Reviewer 3:

Q. The authors have developed a new asymmetric strategy for the vinylation of 1,3-dicarbonyl compounds that merges organocatalysis, photoredox catalysis, and cobalt catalysis. This is clearly an extensive of their previous work (JACS 2022, 144, 10705) that uses a similar ternary catalytic system for the allylation of β -ketocarbonyl compounds. The major difference between this work and the previous work is the selection of alkene. The previous work used α -methylstyrenes whereas this work mostly uses styrenes that lack this methyl group, which ensures that only one elimination pathway is feasible to form the desired products. The substrate scope and limitations have been thoroughly examined and we commend the authors for demonstrating the robustness of their protocol by running the reaction on a gram scale. The mechanistic studies provide some additional insight but again there is some overlap with previous disclosures. Overall, I do not support publication in Nature Communications as this work does not significantly advance upon previous work disclosed by the same authors.

Reply: The referee is certainly right from a certain viewpoint, however, we'd like to point out several of the distinctive features of this work:

- 1) **Dehydrogenative Heck-type coupling as reported in this paper has not been achieved though extensively explored in the literature.** The seemingly simple extension from α -methylstyrene to styrene is nontrivial for the following reasons: internal hydrogen atom transfer, the critical step for the alkenylation process, is disfavored with styrene (see below); on the other hand, the generated radical species easily undergoes addition to another molecule

of styrene leading to undesired byproducts (b1) as experimentally observed; upon extensively screening based on the mechanistic analysis, eventually,

- 2) We developed a **distinctive H-transfer system** involving electron-deficient cobalt catalyst and a proper H-acceptor, which enables unprecedented enantioselective alkenylation reactions with styrenes and 1,1;

3) Our systematic mechanistic studies uncover a **delicate e/proton shuttle** in this triple synergistic system that help to strengthen our understanding of the developed methodologies and free radical chemistry in general.

Overall, we are largely in disagreement with this referee, and the above highlighting points warrant a publication in this journal.

REVIEWERS' COMMENTS

Reviewer #1 (Remarks to the Author):

After the comments have been carefully addressed by the authors, at this stage, I recommend publishing this manuscript in Nature Communications.